# Associations of Single-Nucleotide Polymorphisms in Slovenian Patients with Acute Central Serous Chorioretinopathy

**DOI:** 10.3390/genes13010055

**Published:** 2021-12-25

**Authors:** Peter Kiraly, Andrej Zupan, Alenka Matjašič, Polona Jaki Mekjavić

**Affiliations:** 1Eye Hospital, University Medical Centre Ljubljana, 1000 Ljubljana, Slovenia; peter.kiraly20@gmail.com (P.K.); polona.jaki@guest.arnes.si (P.J.M.); 2Oxford Eye Hospital, Oxford University Hospitals NHS Foundation Trust, Oxford OX3 9DU, UK; 3Institute of Pathology, Faculty of Medicine, University of Ljubljana, 1000 Ljubljana, Slovenia; alenka.matjasic@mf.uni-lj.si; 4Faculty of Medicine, University of Ljubljana, 1000 Ljubljana, Slovenia; 5Institute Jožef Stefan, 1000 Ljubljana, Slovenia

**Keywords:** central serous chorioretinopathy, CSC, genotype–phenotype correlation, collagen, COL4A3, COL4A4, CFH, rs1329428, TNFRSF10A, CDH5

## Abstract

Central serous chorioretinopathy (CSC) is a chorioretinal disease that usually affects the middle-aged population and is characterised by a thickened choroid, retinal pigment epithelium detachment, and subretinal fluid with a tendency towards spontaneous resolution. We investigated 13 single-nucleotide polymorphisms (SNPs) in 50 Slovenian acute CSC patients and 71 healthy controls in Complement Factor H (CFH), Nuclear Receptor Subfamily 3 Group C Member 2 (NR3C2), Cadherin 5 (CDH5) Age-Related Maculopathy Susceptibility 2 (ARMS2), TNF Receptor Superfamily Member 10a (TNFRSF10A), collagen IV alpha 3 (COL4A3) and collagen IV alpha 4 (COL4A4) genes using high-resolution melt analysis. Statistical calculations revealed significant differences in genotype frequencies for CFH rs1329428 (*p* = 0.042) between investigated groups and an increased risk for CSC in patients with TC (*p* = 0.040) and TT (*p* = 0.034) genotype. Genotype–phenotype correlation analysis revealed that CSC patients with CC genotype in CFH rs3753394 showed a higher tendency for spontaneous CSC episode resolution at 3 months from the disease onset (*p* = 0.0078), which could indicate clinical significance of SNP testing in CSC patients. Bioinformatics analysis of the non-coding polymorphisms showed alterations in transcription factor binding motifs for CFH rs3753394, CDH5 rs7499886 and TNFRSF10A rs13278062. No association of collagen IV polymorphisms with CSC was found in this study.

## 1. Introduction

Central serous chorioretinopathy (CSC) is a chorioretinal disease that usually affects the middle-aged population, with a male predilection. It is characterised by a thickened choroid, retinal pigment epithelium (RPE) irregularities or detachment (PED), and subretinal neurosensory retinal detachment due to subretinal fluid (SRF) accumulation [1]. As Daruich et al. showed, it usually follows a self-limiting natural course, with 84% of patients having total SRF reabsorption at 6 months [2]. In chronic CSC, prolonged subretinal fluid accumulation induces hypoxic damage to the outer retina, which results in impaired visual acuity [3].

CSC is related to several risk factors, including arterial hypertension, cardiovascular diseases, Helicobacter pylori infection, gastroesophageal disorders, autoimmune diseases, obstructive sleep apnoea, corticosteroid or sympathomimetic drug usage, type A personality, psychological stress, pregnancy, and alcohol consumption [1,4]. There are few reports of familial CSC occurrence [5,6,7,8]. A small study, in which 16 relatives of CSC patients were observed, showed that 50% of relatives had a thickened choroid (395 mm threshold), which could indicate a possible dominant pattern of transmission [9]. Furthermore, a recent study of 103 relatives of CSC patients from 23 families confirmed familial CSC occurrence. It showed that 44% of relatives had SRF accumulation on optical coherence tomography (OCT) and/or leakage on fluorescein angiography (FA), and an additional 26% had signs suggestive of CSC [10]. Familial CSC occurrence would suggest genetic susceptibility to disease occurrence.

There have been several studies associating CSC with single-nucleotide polymorphisms (SNP) in the complement factor H (*CFH*) gene, age-related maculopathy susceptibility 2 (*ARMS2*) gene, mineralocorticoid receptor encoding (*NR3C2*) gene, cadherin 5 (*CDH5*) gene, and TNF Receptor Superfamily Member 10a (*TNFRSF10A-LOC389641)* gene [11,12,13,14,15,16,17]. CFH is a serum-binding protein that binds to adrenomedullin, which increases choroidal blood flow and is a potent vasodilator, which could influence choroidal thickness in CSC [18,19]. Moreover, the CFH protein, which could be found on a cell surface in the RPE-choroid complex, is an inhibitor of the complement system through inhibition of C3 activity [17]. There have been several *CFH* gene SNPs associated with CSC, which either increased or decreased the risk of disease occurrence [11,13,14]. Apart from *CFH* gene polymorphisms, *ARMS2* gene SNP rs10490924 is associated with CSC, which suggests genetic overlap between age related macular degeneration (AMD) and CSC [11]. Mineralocorticoid receptors (MR) were found in choroid, RPE and neurosensory retina in rats. Inappropriate MR activation induced by intravenous corticosteroid injection caused choroidal enlargement and leakage [20]. Van Dijk et al. showed an association between MR-encoding *NR3C2* gene (rs2070951) and chronic CSC [16]. Downregulation of *CDH5* in choroidal vessels due to corticosteroid intake could cause choroidal vasculature leakage. A study has shown two SNPs in the *CDH5* encoding gene strongly associated with CSC [15]. A recent genome-wide association study (GWAS) identified a new susceptibility gene *TNFRSF10A-LOC389641* to be associated with AMD [12].

Only a few published studies have investigated genotype–phenotype correlations in patients with CSC. Mohabati et al. showed that different CSC phenotypes (acute, non-severe chronic, and severe chronic) had a similar genetic predisposition [21]. However, in the study of Hosoda, time to spontaneous CSC episode resolution differed significantly in patients with different *CFH* I62V genotypes [22]. In another study, Hosoda et al. showed that the A allele of *CFH*-rs800292 conferred the risk for a thicker choroid and CSC development [23]. Moreover, Cho et al. found different SNPs (CFH gene) in CSC patients with and without irregular PED [24].

Two case reports have described a possible association between keratoconus and CSC [25,26]. In keratoconus, which is a basement membrane disease, significant downregulation of genes encoding fibronectin, laminin, and collagen type IV has been reported [26]. Early changes in non-scarring corneal structures (basement membrane, Bowman’s layer) lead to secondary corneal oedema, thinning and scarring [26,27]. Similarly, genetic alterations in CSC patient’s Bruch’s membrane, which contains collagen type IV, could compromise outer blood–retinal barrier and lead to accumulation of SRF.

Genetic studies in CSC patients have mainly been performed on various Western European, Greek, and East Asian populations [11,12,16,21,22,23,24,28,29,30,31]. To date, no genetic study has been performed on Eastern European and Slovenian CSC populations. The aim of our study was to evaluate nine different SNPs in acute CSC Slovenian patients that are already known to be associated with CSC in other populations. Genotypic and phenotypic correlations for these SNPs, such as spontaneous CSC episode resolution and choroidal thickness, were studied. In addition, potential associations between four collagen IV polymorphisms, described in keratoconus, and CSC pathology were studied. Finally, effects of SNPs that are located in the non-coding regions of the investigated genes were analysed with an in silico analysis.

## 2. Materials and Methods

### 2.1. Patient Selection

The study was prospective and conducted at the University Eye Hospital in Ljubljana and at the Institute of Pathology, Faculty of Medicine, University of Ljubljana, between 2018 and 2021. It was approved by the National Medical Ethics Committee at the Ministry of Health (Republic of Slovenia) with ethical approval code 0120-141/2018/4, and conformed to the guidelines of the Declaration of Helsinki.

Consecutive patients diagnosed with acute CSC, who provided informed consent to participate in the study, were enrolled. CSC was defined as a neurosensory retinal detachment due to SRF accumulation, confirmed with spectral domain OCT (SD-OCT) with active fluid leakage on fundus angiography. Exclusion criteria were: CSC episode duration longer than 3 months, choroidal neovascularisation, polypoid choroidal vasculopathy, ocular or systemic diseases that could cause similar changes to the retina, and history of pre-existing macular disease other than CSC. For a control group, healthy blood donors were used with no history of CSC. The control group was sex and age matched with the CSC group. Overall, the study group consisted of 50 acute CSC patients and 71 controls.

### 2.2. Ophthalmological Exam

Patients were examined at baseline and 3 months after the disease onset. At baseline, detailed medical history was taken and patients underwent ophthalmological examination with dilated pupils. Additionally, fundus angiography (fluorescein—FA and indocyanine green—ICG), fundus autofluorescence (FAF) and SD-OCT of the macula (Spectralis, Heidelberg Engineering, Inc., Heidelberg, Germany) were performed to confirm the diagnosis (Figure 1A,B).

Enhanced depth imaging (EDI) SD-OCT was performed in the central 30° of retina to determine subretinal fluid accumulation and thickness of the choroid (Figure 1C,D). The vertical distance from the outermost part of the Bruch’s membrane to the sclerochoroidal junction was measured to determine choroidal thickness at five different locations (under the foveola, 500 and 1000 microns nasally and temporally from the foveola). Average of these measurements was calculated and used as the choroidal thickness (Figure 1E).

The course of the disease was monitored by SD-OCT. Patients were divided into two groups according to the SRF persistence on the follow-up (3 months after disease onset). Patients in the spontaneous resolution group did not have any SRF 3000 microns around the fovea, while patients in the persistent group had persistent SRF.

### 2.3. Sample Collection for Genetic Testing and DNA Extraction

All patients and controls signed informed consent for genetic testing, in which the study management was described. Genomic DNA was extracted from whole blood using an automated paramagnetic particles-based isolation kit (Maxwell RSC Whole Blood DNA Kit, Promega Corporation, Madison, WS, USA). DNA concentration and quality were determined spectrophotometrically using Nanodrop One (Thermo Fisher Scientific, Wilmington, DE, USA).

### 2.4. DNA Amplification and Genotyping

For the purpose of this study, 13 SNPs; rs5522 (*NR3C2)*, rs1329428 (*CFH)*, rs3753394 (*CFH)*, rs1065489 (*CFH)*, rs7499886 (*CDH5*), rs800292 (*CFH)*, rs10490924 (*ARMS2)*, rs13278062 (*TNFRSF10A)*, rs1061170 (*CFH)*, rs10178458 (COL4A3), rs55703767 (COL4A3), rs2229814 (COL4A4) and rs2229813 (COL4A4) were selected (Table 1). The selection of the specific group of SNPs was based on previous research and their specific relationship with CSC phenotypes and keratoconus, as identified mostly with GWAS [11,12,15,16,21,22,23,30,32].

PCR amplification of all SNPs was performed with genomic DNA with one set of primers, as described previously (Appendix A) [33]. In brief, all PCRs were performed in a 15 μL reaction mixture containing 30 ng of purified genomic DNA, 200 nM of each of the primers and 1 × Type-It HRM PCR Kit (Qiagen, Hilden, Germany). PCR reactions were optimised: 5 min at 95 °C, followed by 45 cycles as follows: 10 s at 95 °C, 30 s at 56 °C and 10 s at 72 °C, with detection of the fluorescence on the Green channel. Melting temperature was raised from 65 to 95 °C with a ramp of 0.02 °C per second and detection of fluorescence on the HRM channel. Rotor-Gene Q 5plex HRM was used to perform the reactions (Qiagen, Hilden, Germany). Analysis of the HRM results was conducted using Rotor-Gene Q Series software, 2.0.2 (Qiagen, Hilden, Germany). To discriminate between different genotypes, HRM analysis was performed as follows: the melt plots were normalised and then transformed to difference plots, as a representation of the difference in fluorescence between samples to a selected control at each temperature transition (Appendix A). In order to produce a normalised melting graph, two normalised regions were selected: one encompassing the representative baseline data for the pre-melt phase and the other encompassing representative data for the post-melt phase. In case of ambiguous genotypes obtained by HRM, further validation of results was performed by Sanger sequencing using standard BigDye 3.1 chemistry and SeqStudio instrument (both from Thermo Fisher Scientific, Madison, WI, USA).

### 2.5. Statistical Analysis

The case and control populations were tested for the Hardy–Weinberg equilibrium (HWE). Statistical differences in SNPs genotype distribution were tested by use of a chi-square test with estimation of the odds ratios (OR) for each genotype in respect to the reference genotype. Estimated OR were obtained with 95% confidence intervals and all statistical tests were considered bilateral with a significance level of 0.05. Haplotypes were analysed with an Expectation Maximisation (EM) algorithm and associations between haplotypes and disease were analysed via logistic regression. The HWE test, chi-square test with estimation of OR and a likelihood ratio and haplotype analysis were performed using SNP Stats software [34].

### 2.6. Transcriptional Factors Binding Sites Analysis

The influence of SNPs in non-coding and untranslated (UTRs) regions on transcription factor binding sites (TFBS) was determined using Promo software version 3.0.2 (Alggen) [35]. Promo is using Transfac 8.3 database to construct specific binding site weight matrices for TFBS prediction. Only human transcription sites and factors were considered for the calculations.

## 3. Results

### 3.1. Phenotypic Characteristics of the CSC Patients

Patient characteristics are presented in Table 2. The cohort of 50 CSC patients included 43 men (86%) and 7 women (14%). Three months after the disease onset, 19 (38%) patients had spontaneous CSC episode resolution and 31 (62%) patients had persistent CSC episodes. The average choroidal thickness at baseline was 446 ± 116 μm.

### 3.2. Genotypes Frequencies for the Analysed SNPs

Among the 13 investigated SNPs, all analysed SNPs were determined to be in Hardy–Weinberg equilibrium (*p* > 0.05) for both the case and control group, with the exception of rs1061170 and rs13278062 (Table 3). In case of rs1061170, both control (*p* = 0.037) and CSC (*p* = 0.012) group were in HWE disequilibrium, and in case of rs13278062, the control group (*p* = 0.031) was in HWE disequilibrium. Differences between genotypes of the CSC and control group revealed significant values for rs1329428 (CFH) (*p* = 0.042) and rs1061170 (CFH) (*p* = 0.013) (Table 3).

### 3.3. Impact of Analysed SNPs for the CSC Development

Odds ratio calculations revealed an increased risk for the CSC development in patients with TC (OR = 2.89, 95% CI = 1.08–7.75, *p* = 0.040) and TT (OR = 3.66, 95% CI = 1.22–10.96, *p* = 0.034) genotype in comparison to CC genotype in case of rs1329428 (CFH) (Table 4). Furthermore, analysis revealed a protective role of the homozygous TT genotype (OR = 0.14, 95% CI = 0.02–1.14, *p* = 0.048) for the rs1061170 (CFH), when compared to CC reference genotype; however, rs1061170 was found to be in HWE disequilibrium.

### 3.4. Impact of Analysed SNPs for the CSC Phenotype Modulation

In order to evaluate the role of the investigated SNPs in modulating the phenotype in CSC patients, associations between phenotypic characteristics and genotypes were calculated. In the case of rs3753394 (CFH), patients with the CC genotype showed a higher tendency for spontaneous resolution when compared to patients with CT and TT genotypes (OR = 5.09; 95% CI = 1.45–17.92, *p* = 0.0078). When investigating associations between SNPs and choroidal thickness, no significant associations were detected.

### 3.5. Analysis of Association between Haplotypes and Disease

To determine possible relations between collagen genes polymorphisms and the risk of CSC development, haplotypes based on four investigated collagen SNPs were constructed, and an odds ratio between reference haplotype (CGTA) and constructed haplotypes was determined. Analysis revealed no significant values between specific haplotypes and the risk for CSC development (Table 5).

### 3.6. Transcriptional Factor Binding Sites Predictions

To assess the effects of polymorphisms in the non-coding regions of the investigated genes for the binding of transcription factors, we performed an in silico analysis of possible binding sites and effects of nucleotide changes (Table 6). The analysis revealed that the presence of T allele in rs3753394 in the upstream location of the CFH gene deletes binding sites for four different transcription factors: TFII-I, GR-alpha, GATA-1, and T3R-beta1. On the other hand, the allelic change G to T of the rs13278062 upstream of the TNFRSF10A gene creates a binding site for GR-alpha. As for rs7499886 in the intron region of the CDH5, our analysis showed that the A allele is associated with the GR-beta transcription factor, whereas the G allele is associated with ENKTF-1 transcription factor. In the case of rs1329428, no differences were observed in transcription factor binding site motif.

## 4. Discussion

The aim of our study was to assess association between 13 SNPs and a risk of CSC in the Slovenian population. Furthermore, genotype–phenotype correlations were calculated, and in silico analysis was performed to evaluate effects of SNPs in the non-coding regions of the investigated genes.

Studies have revealed higher CSC incidences in some ethnic groups; however, a relatively small number of studies have been conducted to evaluate genetic differences in CSC populations, which could contribute to different CSC incidences [1]. Out of nine SNPs that were associated with CSC patients in other studies, only rs1329428 was found to be associated with Slovenian CSC patients. No association between specific collagen IV alpha 3 and alpha 4 SNPs (rs10178458, rs55703767, rs2229814, rs2229813) and CSC patients was found in our study. We have found previously undescribed genotype–phenotype association between rs3753394 and spontaneous CSC episode resolution. Moreover, bioinformatic analysis revealed previously undescribed effects of polymorphisms in the non-coding regions of the investigated genes for the binding of transcription factors.

The *CFH* gene encodes complement factor H (FH) protein, which acts as an inhibitor of the complement system by blocking the formation of C3-converstases [36]. We examined five SNPs in *CFH* gene and after Chi-square and Fisher exact test were performed, rs1329428 and rs1061170 were observed to have significant differences in genotypes frequencies between CSC patients and control group, however, rs1061170 was found to be in Hardy–Weinberg disequilibrium in both case and control group. Odds ratio analysis revealed an increased risk of the rs1329428 TC and TT genotypes for the CSC development. Of all CFH SNPs found to be associated with CSC, the strongest association was observed with rs1329428 (TC genotype), which is in concordance with the results of our study [13]. In a study by Yoneyama et al., the minor T-allele of rs1329428 was associated with increased choroidal thickness and vascular permeability in patients with polypoidal choroidal vasculopathy [37]. No genotype–phenotype correlation between CSC and rs1329428 was observed in our study. Although it has been suggested that rs1329428 can bind to transcriptional regulatory proteins, the eQTL expression data do not support that finding [14].

Collagen is the main component of Bruch’s membrane; specifically, the retinal pigment epithelium basal lamina contains collagen IV α3–5 [38]. Suppression of collagen synthesis may negatively influence any reparative process in the damaged choroidal circulation leading to a persistent choroidal leakage. Furthermore, due to collagen changes in the Bruch’s membrane, the outer blood–retinal barrier could be compromised. The investigated polymorphisms in the collagen IV genes (rs10178458, rs55703767, rs2229814, rs2229813) in this study did not reveal any significant differences between genotypes of the CSC and control group. Additionally, odds ratio calculation did not show an increased risk in the presence of the above-mentioned SNPs. To assess a combined effect of collagen SNPs, haplotypes were constructed and risk assessment was determined for specific haplotypes, which did not reveal any significant values. Our study focused on four collagen IV SNPs, associations of which were previously described in patients with keratoconus [32]. These associations have not been shown in CSC patients, but this result should not exclude the role of collagen genes in CSC pathology, since other collagen type IV polymorphisms may be implicated.

Results of our study showed a strong association (*p* = 0.0078) between the CC genotype in rs3753394, which is located in the promotor region of the CFH gene, and the spontaneous CSC episode resolution 3 months after the disease onset. The *CFH*-rs3753394 was determined as a risk factor for CSC in the Japanese and Dutch populations and as a protective factor for CSC in the Greek population [11,13,14]. In a study by Hosoda et al., *CFH*-rs3753394 was observed to be associated with choroidal thickness [23], whereas no significant association between rs3753394 and choroidal thickness was found in our study population. The C allele was considered to be an allele of effect in a study of choroidal thickness, so the association between choroidal thickness and spontaneous CSC episode resolution should be confirmed by future studies [23].

Although the mechanism of *CFH* involvement in the development of CSC is still unknown, the specific locations of many CSC-associated SNPs in the promoter and intron regions of the *CFH* gene suggest a regulatory role, which may affect expression of the gene in a specific tissue. Our bioinformatics analysis of the non-coding SNPs (rs1329428, rs3753394, rs7499886, rs13278062) revealed significant alterations in transcriptional binding factors. Although we did not detect any change in transcriptional factors binding motifs of the rs1329428, loss of binding sites for the transcription factors TFII-I, GR-alpha, GATA-1, and T3R-beta1 was detected in case of rs3753394. rs3753394 is located in the CFH promoter region, and the presence of T alleles may impair the promoter function, thus regulating the CFH gene expression. In silico analysis of the rs7499886 revealed deletion of the binding motif for GR-beta and creation of a binding motif for ENKTF-1 transcription factor. rs7499886 is located in the intron region of the CDH5 gene, which was suggested to be involved in the CSC disease mechanism due to its function in endothelial cell–cell junctions, and its localisation in the choriocapillaris [15]. In the case of rs13278062, we detected a novel binding site for GR-alpha. rs13278062 is located upstream of the TNFRSF10A gene. TNFRSF10A was first identified as an AMD susceptibility locus in the Japanese population [39] and has since been confirmed in other ethnic groups [40,41,42]. However, the exact role of TNFRSF10A in CSC pathology is unclear, but since eQTL of rs13278062 has been shown to be associated with its expression in the adrenal glands, TNFRSF10A may affect the risk of CSC by modulating hormone secretion from the adrenal glands [12].

## 5. Conclusions

In conclusion, our study demonstrated a significantly increased risk for Slovenian patients with acute CSC carrying specific *CFH* genotypes for the first time, and also assessed potential associations between collagen IV gene polymorphisms in CSC patients for the first time. Moreover, we have found previously undescribed genotype–phenotype correlations in CSC patients, and described the effects of polymorphisms in the non-coding regions of the investigated genes for the binding of transcription factors.

## Figures and Tables

**Figure 1 genes-13-00055-f001:**
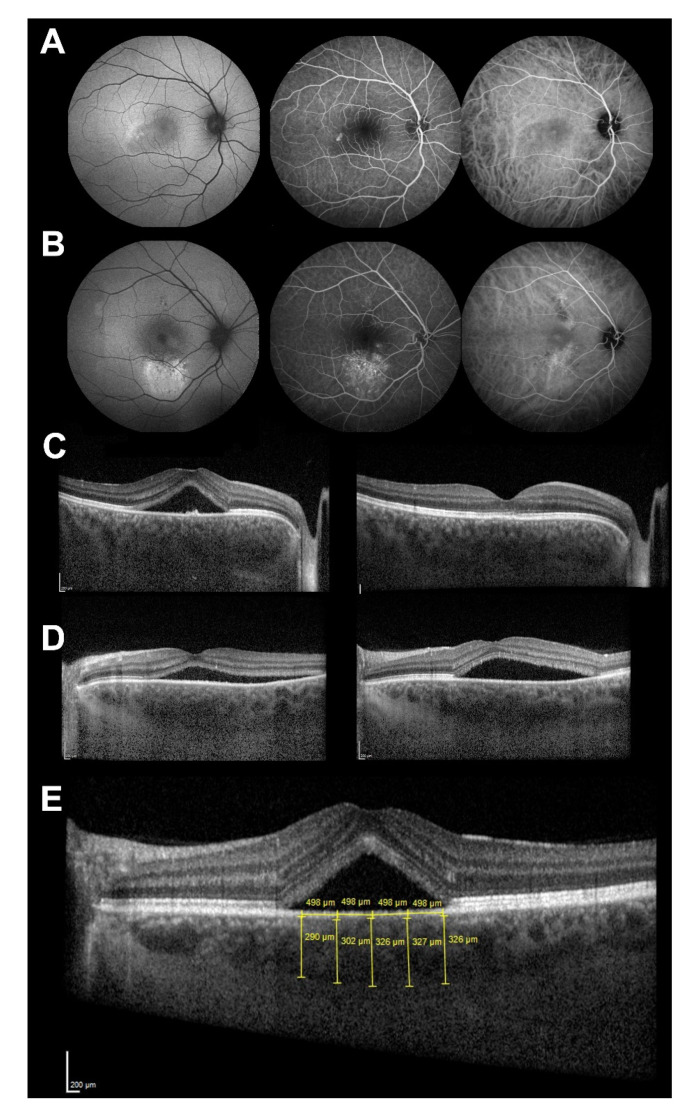
(**A**,**B**) Central serous chorioretinopathy (CSC) clinical features on multimodal imaging to confirm the diagnosis in two patients; (**C**) optical coherence tomography (OCT) of a patient with spontaneous CSC episode resolution; at baseline (left) and 3 months thereafter (right); (**D**) OCT of a patient with persistent CSC episode; at baseline (left) and 3 months thereafter (right); (**E**) Measurement of choroidal thickness with enhanced depth imaging OCT. Average of five measurements was calculated and used as the choroidal thickness.

**Table 1 genes-13-00055-t001:** Description of the investigated single-nucleotide polymorphisms (SNPs) with minor frequencies. MAF: minor allelic frequency; Total: global frequencies across all populations as obtained from dbSNP database (https://www.ncbi.nlm.nih.gov/snp/ (accessed on the 25 October 2021); European: European frequencies as obtained from the dbSNP database; CSC: central serous chorioretinopathy.

SNP	Gene	Published Associationswith CSC	Variant Type	Genomic Location (GRCh38.p13)	MAF			
					Total	European	CSC Patients	Control Group
rs10490924	*ARMS2*	Yes	Missense	chr10:122454932	0.245	0.238	0.250	0.239
rs7499886	*CDH5*	Yes	Intron	chr16:66379292	0.443	0.431	0.470	0.451
rs1329428	*CFH*	Yes	Intron	chr1:196733680	0.404	0.395	0.410	0.563
rs3753394	*CFH*	Yes	Upstream	chr1:196651787	0.286	0.288	0.290	0.246
rs1065489	*CFH*	Yes	Missense	chr1:196740644	0.169	0.171	0.160	0.127
rs800292	*CFH*	Yes	Missense	chr1:196673103	0.248	0.222	0.340	0.254
rs1061170	*CFH*	Yes	Missense	chr1:196690107	0.371	0.376	0.310	0.346
rs10178458	*COL4A3*	No	Missense	chr2:227246719	0.173	0.168	0.170	0.157
rs55703767	*COL4A3*	No	Missense	chr2:227256385	0.204	0.218	0.260	0.300
rs2229814	*COL4A4*	No	Missense	chr2:227089883	0.499	0.496	0.440	0.514
rs2229813	*COL4A4*	No	Missense	chr2:227028004	0.435	0.427	0.430	0.457
rs5522	*NR3C2*	Yes	Missense	chr4:148436323	0.114	0.110	0.070	0.100
rs13278062	*TNFRSF10A*	Yes	Upstream	chr8:23225458	0.496	0.488	0.380	0.486

**Table 2 genes-13-00055-t002:** Characteristics of investigated acute central serous chorioretinopathy (CSC) patients and healthy controls.

	CSC Patients		Controls	
	N	N (%)	N	N (%)
Total samples	50		71	
Female	7	14%	9	12.7%
Male	43	86%	62	87.3%
Mean age at diagnosis: years (± SD)	44.7 (±10)	/	45.7 (±10)	/
No. of eyes (no., %) with spontaneous CSC episode resolution at 3 months	19	38%	/	/
No. of eyes (no., %) with persistent CSC episode at 3 months	31	62%	/	/
Average choroidal thickness, μm, mean (± SD)	446 (±116)	/	/	/

**Table 3 genes-13-00055-t003:** Genotype frequencies and differences in genotype distribution of 13 investigated single nucleotide polymorphisms (SNPs). CSC: central serous chorioretinopathy patients; HWE: Hardy–Weinberg equilibrium; χ^2^: chi-square statistics. Bold font indicates data considered to be statistically significant (*p* < 0.05).

Genotypes	Control Group			Central Serous Chorioretinopathy Patients				
	N (%)	HWE		N (%)	HWE		CSC	
		χ^2^	*p* Value		χ^2^	*p* Value	χ^2^	*p* Value
**rs5522**	**70**	0.864	0.352	50	0.283	0.594	/	/
T/C	14 (20.0)			7 (14.0)				
T/T	56 (80.0)			43 (86.0)				
C/C	0 (0.0)			0 (0.0)				
**rs1329428**	71	0.499	0.479	50	0.678	0.411	6.32	**0.042**
T/T	15 (21.1)			16 (32.0)				
T/C	32 (45.1)			27 (54.0)				
C/C	24 (33.8)			7 (14.0)				
**rs3753394**	71	2.185	0.139	50	0.020	0.888	1.68	0.432
C/C	38 (53.5)			25 (50.0)				
C/T	31 (43.7)			21 (42.0)				
T/T	2 (2.8)			4 (8.0)				
**rs1065489**	71	0.023	0.879	50	0.574	0.448	0.89	0.642
G/G	54 (76.1)			36 (72.0)				
G/T	16 (22.5)			12 (24.0)				
T/T	1 (1.4)			2 (4.0)				
**rs7499886**	71	0.465	0.495	50	0.352	0.552	0.10	0.953
A/A	20 (28.2)			13 (26.0)				
A/G	38 (53.5)			27 (54.0)				
G/G	13 (18.3)			10 (20.0)				
**rs800292**	71	0.961	0.326	50	0.019	0.889	1.82	0.404
A/A	3 (4.2)			6 (12.0)				
G/A	30 (42.3)			22 (44.0)				
G/G	38 (53.5)			22 (44.0)				
**rs10490924**	71	0.367	0.544	50	0.009	0.924	0.08	0.960
G/G	42 (59.2)			28 (56.0)				
G/T	24 (33.8)			19 (38.0)				
T/T	5 (7.0)			3 (6.0)				
**rs13278062**	70	4.607	**0.031**	50	2.785	0.095	1.47	0.480
G/G	21 (30.0)			10 (20.0)				
T/G	26 (37.1)			18 (36.0)				
T/T	23 (32.9)			22 (44.0)				
**rs1061170**	68	4.326	**0.037**	50	6.329	**0.012**	8.65	**0.013**
C/C	12 (17.6)			1 (2.0)				
T/C	23 (33.8)			29 (58.0)				
T/T	33 (48.5)			20 (40.0)				
**rs10178458**	70	2.433	0.118	50	2.429	0.119	2.79	0.248
C/C	48 (68.6)			36 (72.0)				
C/T	22 (31.4)			11 (22.0)				
T/T	0 (0.0)			3 (6.0)				
**rs55703767**	70	0.548	0.459	50	1.417	0.233	1.57	0.457
G/G	33 (47.1)			29 (58.0)				
G/T	32 (45.7)			16 (32.0)				
T/T	5 (7.1)			5 (10.0)				
**rs2229814**	70	1.447	0.228	50	0.034	0.854	1.55	0.460
C/C	16 (22.9)			10 (20.0)				
T/C	40 (57.1)			24 (48.0)				
T/T	14 (20.0)			16 (32.0)				
**rs2229813**	70	0.032	0.858	50	0.516	0.473	0.32	0.853
A/A	15 (21.4)			8 (16.0)				
G/A	34 (48.6)			27 (54.0)				
G/G	21 (30.0)			15 (30.0)				

**Table 4 genes-13-00055-t004:** Odds ratio (OR) calculations for the investigated single nucleotide polymorphisms (SNPs). CSC: central serous chorioretinopathy; OR: odds ratio. Bold font indicates data considered to be statistically significant (*p* < 0.05).

Genotypes	CSC	
	OR (95% CI)	*p* Value
**rs5522**		
T/C	1.00	
T/T	0.65 (0.24–1.75)	0.470
C/C	/	/
**rs1329428**		
C/C	1.00	
T/C	2.89 (1.08–7.75)	**0.040**
T/T	3.66 (1.22–10.96)	**0.034**
**rs3753394**		
C/C	1.00	
C/T	1.03 (0.49–2.18)	1.000
T/T	3.04 (0.52–17.86)	0.230
**rs1065489**		
G/G	1.00	
G/T	1.12 (0.48–2.66)	0.828
T/T	3.00 (0.26–34.33)	0.565
**rs7499886**		
A/A	1.00	
A/G	1.09 (0.46–2.57)	1.00
G/G	1.18 (0.40–3.49)	0.789
**rs800292**		
G/G	1.00	
G/A	1.27 (0.59–2.71)	0.566
A/A	3.45 (0.78–15.21)	0.144
**rs10490924**		
G/G	1.00	
G/T	1.19 (0.55–2.56)	0.697
T/T	0.90 (0.20–4.07)	1.000
**rs13278062**		
G/G	1.00	
T/G	0.72 (0.31–1.67)	0.525
T/T	0.50 (0.19–1.29)	0.165
**rs1061170**		
T/T	1.00	
T/C	2.08 (0.95–4.54)	0.0794
C/C	0.14 (0.02–1.14)	**0.0477**
**rs10178458**		
C/C	1.00	
C/T	0.67 (0.29–1.55)	0.405
T/T	/	/
**rs55703767**		
G/G	1.00	
G/T	0.57 (0.26–1.24)	0.175
T/T	1.14 (0.30–4.33)	1.000
**rs2229814**		
C/C	1.00	
T/C	0.53 (0.22–1.26)	0.182
T/T	0.55 (0.19–1.59)	0.295
**rs2229813**		
A/A	1.00	
G/A	1.11 (0.48–2.56)	0.835
G/G	0.75 (0.25–2.21)	0.785

**Table 5 genes-13-00055-t005:** Odds ratio (OR) calculations of the collagen genes single-nucleotide polymorphisms (SNPs) haplotypes. Haplotypes were constructed based on collagen IV genes’ polymorphisms.

	rs10178458	rs55703767	rs2229814	rs2229813	Freq	OR (95% CI)	*p* Value
1	C	G	T	A	0.2876	1.00	---
2	C	G	C	G	0.2502	1.58 (0.66–3.77)	0.31
3	C	T	C	G	0.1378	0.39 (0.11–1.35)	0.14
4	C	T	T	G	0.0498	2.33 (0.47–11.65)	0.3
5	T	G	T	A	0.0438	3.75 (0.44–31.79)	0.23
6	C	G	T	G	0.0408	1.69 (0.27–10.41)	0.57
7	T	G	C	G	0.0360	0.29 (0.03–2.99)	0.3
8	T	T	T	A	0.0279	3.71 (0.39–34.99)	0.25
9	C	T	T	A	0.0272	1.03 (0.05–22.93)	0.99
10	T	G	T	G	0.0266	0.50 (0.05–5.42)	0.57
11	C	G	C	A	0.0242	0.37 (0.02–7.16)	0.51
12	C	T	C	A	0.0199	1.59 (0.14–17.88)	0.71
13	T	T	T	G	0.0130	4.10 (0.05–309.99)	0.52
rare	*	*	*	*	0.0152	/	/

* other possible haplotypes.

**Table 6 genes-13-00055-t006:** Transcription factor binding sites predictions for the studied non-coding polymorphisms using PROMO (ALLGEN) software. Changes in transcriptional factor binding sites are indicated in bold letters.

SNP (Gene)	rs1329428 (CFH)		rs3753394 (CFH)		rs7499886 (CDH5)		rs13278062 (TNFRSF10A)	
Alleles	T	C	C	T	A	G	G	T
Transcription factors	TFIID	TFIID	**TFII-I**	/	**GR-beta**	**ENKTF-1**	/	**GR-alpha**
			**GR-alpha**	/				
	GR-beta	GR-beta	**GATA-1**	/				
			**T3R-beta1**	/				

## Data Availability

Data available on request.

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
