# Peer review of "Associations of Single-Nucleotide Polymorphisms in Slovenian Patients with Acute Central Serous Chorioretinopathy"

_genes, 2021, doi:10.3390/genes13010055_

Round 1

Reviewer 1 Report

A study by Peter Kiraly et al. (Manuscript ID: genes-1513976) presenting associations of single nucleotide polymorphisms in 50 patients with acute central serous chorioretinopathy in the Slovenian population.

I would like to congratulate the authors on this interesting study, presented in a very clear way. This is an interesting work, where genetic analysis can assist in assessing biomarkers for disease prognosis and management in this specific Slovenian population, and may have an insight into studies of e.g., other similar European populations.

This study has few main points:

  1. The study confirms that genetic analysis can assist in the management of patients with CSC.
  2. Despite being focused on a small Slovenian population, this work is novel and shows for the first time increased risk for acute CSC in patients carrying specific CFH genotypes. Furthermore, it points out the association of the collagen IV gene polymorphisms in CSC patients, which was not described before.
  3. The manuscript is very well-written and clear.
  4. Images and tables are of very good quality and clear.

Weaknesses:

  1. Minor language improvements should be addressed.

Broad comments:

  1. Perhaps highlight in the abstract about clinical significance of SNPs testing.
  2. The discussion Section, row 311, should start with some form of introduction (two sentences) about CSC and the aim of the study.

Specific comments:

  1. Row 118-119: remove additional spacing.
  2. Add colon after the word ‘were’ in row 135.
  3. Row 154: change to the Bruch’s membrane.

Author Response

Dear reviewer,

Best wishes,

Peter Kiraly

Reviewer 2 Report

Journal “Genes”

Review comments:

Title of the manuscript: Association of Single nucleotide polymorphism in Slovenian patients with acute central serous chorioretinopathy.

This is an interesting study investigated the “central serous chorioretinopathy” (CSC), a recurrent chorioretinal disease that affects the middle-aged populations. In this study, authors investigated 13 SNPs in 50 different Slovenian patients with adequate amounts of healthy controls. Genotype-phenotype correlations revealed that higher tendency for spontaneous CSC episode resolution at 3 months from the disease onset. Bioinformatics analysis showed alterations in CFH, CDH5, and TNFRSF10A. However, it was unclear about the statement that collagen 4 polymorphisms with CSC pathology were not established. Overall, there are few issues need to be addressed as described below.

1) The authors claimed that 13 SNPs were studied in the abstract (lines 19-21), while the aim of the study was intended to evaluate 9 different SNPs in Slovenian patients with acute SCC in the introduction (lines 133-116). How many SNPs were actually studied in this study?

2) The authors mentioned that the association of collagen IV polymorphisms with the pathology of SCC was not established in this study (lines 34 & 35), whereas the observation presented in the introductory section has a contradictory statement (lines 119-120). What is the exact association status of COL4A3 & COL4A4? Rephrase this statement and to be clear!

3) The study used a high-resolution melt (HRM) approach to genotype the study populations. It is a common approach that HRM is used in the first phase to screen populations based on the melting properties of the targeted region (amplicon). The screened population will be confirmed by standard genotyping approaches (e.g., TaqMan genotyping assays, DNA sequencing, etc.). As many thermodynamic factors will influence the melting properties of the amplicon, the authors can briefly explain the HRM genotyping when they used a special HRM assay with SNP-specific probe melting approach - which might be a better approach than the conventional whole amplicon melting approach.

4) The authors should consider including the list of the HRM primers used in this study, the sizes of amplicons targeted in HRM genotyping, and the single nucleotide base change associated with the melting. The HRM differential melting curve can also be considered.   

Overall, I find this study is interesting and need to address above comments.

Author Response

(The authors gave the same response as above.)
